# STOCHASTIC CANONICAL CORRELATION ANALYSIS: A RIEMANNIAN APPROACH

## ABSTRACT

We present an efficient stochastic algorithm (RSG+) for canonical correlation analysis (CCA) derived via a differential geometric perspective of the underlying optimization task. We show that exploiting the Riemannian structure of the problem reveals natural strategies for modified forms of manifold stochastic gradient descent schemes that have been variously used in the literature for numerical optimization on manifolds. Our developments complement existing methods for this problem which either require $O(d^3)$ time complexity per iteration with $O(\frac{1}{\sqrt{t}})$ convergence rate (where $d$ is the dimensionality) or only extract the top 1 component with $O(\frac{1}{t})$ convergence rate. In contrast, our algorithm achieves $O(d^2k)$ runtime complexity per iteration for extracting top $k$ canonical components with $O(\frac{1}{t})$ convergence rate. We present our theoretical analysis as well as experiments describing the empirical behavior of our algorithm, including a potential application of this idea for training fair models where the label of protected attribute is missing or otherwise unavailable.

## 1 INTRODUCTION

Canonical correlation analysis (CCA) is a popular method for evaluating correlations between two sets of variables. It is commonly used in unsupervised multi-view learning, where the multiple views of the data may correspond to image, text, audio and so on Rupnik & Shawe-Taylor (2010); Chaudhuri et al. (2009); Luo et al. (2015). Classical CCA formulations have also been extended to leverage advances in representation learning, for example, Andrew et al. (2013) showed how the CCA can be interfaced with deep neural networks enabling modern use cases. Many results over the last few years have used CCA or its variants for problems including measuring representational similarity in deep neural networks Morcos et al. (2018), speech recognition Couture et al. (2019), etc.

The goal in CCA is to find linear combinations within two random variables $\mathbf{X}$ and $\mathbf{Y}$ which have maximum correlation with each other. Formally, the CCA problem is defined in the following way. Given a pair of random variables, a $d_x$-variate random variable $\mathbf{X}$ and a $d_y$-variate random variable $\mathbf{Y}$, with unknown joint probability distribution, find the projection matrices $U \in \mathbf{R}^{d_x \times k}$ and $V \in \mathbf{R}^{d_y \times k}$, with $k \leq \min\{d_x, d_y\}$, such that the correlation is maximized:

$$\text{maximize trace} \left( U^T \mathbb{E}_{\mathbf{X},\mathbf{Y}} \left[ X^T Y \right] V \right) \text{ s.t. } U^T \mathbb{E}_{\mathbf{X}} \left[ X^T X \right] U = I_k, V^T \mathbb{E}_{\mathbf{Y}} \left[ Y^T Y \right] V = I_k \quad (1)$$

Here, $X, Y$ are samples of $\mathbf{X}$ and $\mathbf{Y}$ respectively. The objective function in (1) is the expected cross-correlation in the projected space and the constraints specify that different canonical components should be decorrelated. Let $C_X = \mathbb{E}_{\mathbf{X}}[X^T X]$ and $C_Y = \mathbb{E}_{\mathbf{Y}}[Y^T Y]$ be the covariance matrices, and $C_{XY} = \mathbb{E}_{(\mathbf{X},\mathbf{Y})}[X^T Y]$ denote cross-covariance. Let us define the whitened covariance $T :=$ $C_X^{-1/2} C_{XY} C_Y^{-1/2}$ and $\Phi_k$ (and $\Psi_k$) contains the top-$k$ left (and right) singular vectors of $T$. It is known Golub & Zha (1992) that the optimum of (1) is achieved at $U^* = C_X^{-1/2} \Phi_k$, $V^* = C_Y^{-1/2} \Psi_k$.

In practice, we may be given two views of $N$ samples as $X \in \mathbf{R}^{N \times d_x}$ and $Y \in \mathbf{R}^{N \times d_y}$. A natural approach to solving CCA is based on on the following sequence of steps. We first compute the *empirical* covariance and cross-covariance matrices, namely, $\widetilde{C}_{\mathbf{X}} = 1/N X^T X$, $\widetilde{C}_{\mathbf{Y}} = 1/N Y^T Y$ and $\widetilde{C}_{\mathbf{XY}} = 1/N X^T Y$. We then calculate the empirical whitened cross-covariance matrix $\widetilde{T}$, finally, compute $U^*, V^*$ by applying a $k$-truncated SVD to $\widetilde{T}$.

**Runtime and memory considerations.** The above procedure is simple but is only feasible when the data matrices are small. But in most modern applications, not only are the datasets large but also the dimension $d$ (let $d = \max\{d_x, d_y\}$) of each sample can be quite high, especially if representations are being learned using deep neural network models. As a result, the computational footprint of the foregoing algorithm can be quite high. This has motivated the study of stochastic optimization routines for solving CCA. Observe that in contrast to the typical settings where stochastic optimization schemes are most effective, the CCA objective does not decompose over samples in the dataset. Many efficient strategies have been proposed in the literature: for example, Ge et al. (2016); Wang et al. (2016) present Empirical Risk Minimization (ERM) models which optimize the empirical objective. More recently, Gao et al. (2019); Bhatia et al. (2018); Arora et al. (2017) describe proposals that optimize the population objective. To summarize the approaches succinctly, if we are satisfied with identifying the top 1 component of CCA, effective schemes are available by utilizing either extensions of the Oja's rule Oja (1982) to generalized eigenvalue problem Bhatia et al. (2018) or the alternating SVRG algorithm Gao et al. (2019)). Otherwise, a stochastic approach must make use of an explicit whitening operation which involves a cost of $d^3$ for each iteration Arora et al. (2017).

**Observation.** Most approaches either directly optimize (1) or instead a reparametrized or regularized form Ge et al. (2016); Allen-Zhu & Li (2016); Arora et al. (2017). Often, the search space for $U$ and $V$ corresponds to the entire $\mathbf{R}^{d \times k}$ (ignoring the constraints for the moment). But if the formulation could be cast in a form which involved approximately writing $U$ and $V$ as a product of several matrices with nicer properties, we may obtain specialized routines which are tailored to exploit those properties. Such a reformulation is not difficult to derive – where the matrices used to express $U$ and $V$ can be identified as objects that live in well studied geometric spaces. Then, utilizing the geometry of the space and borrowing relevant tools from differential geometry leads to an efficient approximate algorithm for top-$k$ CCA which optimizes the population objective in a streaming fashion.

**Contributions.** (a) First, we re-parameterize the top-$k$ CCA problem as an optimization problem on specific matrix manifolds, and show that it is equivalent to the original formulation in equation 1. (b) Informed by the geometry of the manifold, we derive stochastic gradient descent algorithms for solving the re-parameterized problem with $O(d^2 k)$ cost per iteration and provide convergence rate guarantees. (c) This analysis provides a direct mechanism to obtain an upper bound on the number of iterations needed to guarantee an $\epsilon$ error w.r.t. the population objective for the CCA problem. (d) The algorithm works in a streaming manner so it easily scales to large datasets and we do not need to assume access to the full dataset at the outset. (e) We present empirical evidence for both the standard CCA model and the DeepCCA setting Andrew et al. (2013), describing advantages and limitations.

## 2 STOCHASTIC CCA: REFORMULATION, ALGORITHM AND ANALYSIS

The formulation of Stochastic CCA and the subsequent optimization scheme will seek to utilize the geometry of the feasible set for computational gains. Specifically, we will use the following manifolds (please see Absil et al. (2007) for more details):

(a) Stiefel: $\mathsf{St}(p, n)$. The manifold consists of $n \times p$, with $p < n$, column orthonormal matrices, i.e., $\mathsf{St}(p, n) = \{X \in \mathbf{R}^{n \times p} | X^T X = I_p\}$.
(b) Grassmanian: $\mathsf{Gr}(p, n)$. The manifold consists of $p$-dimensional subspaces in $\mathbf{R}^n$, with $p < n$.
(c) Rotations: $\mathsf{SO}(n)$. the manifold/group consists of $n \times n$ special orthogonal matrices, i.e., $\mathsf{SO}(n) = \{X \in \mathbf{R}^{n \times n} | X^T X = X X^T = I_n, \det(X) = 1\}$.

We summarize certain geometric properties/operations for these manifolds in the Appendix but have been leveraged in recent works for other problems also Li et al. (2020); Rezende et al. (2020).

Let us recall the objective function for CCA as given in (1). We denote $X \in \mathbf{R}^{N \times d_x}$ as the matrix consisting of the samples $\{\mathbf{x}_i\}$ drawn from a zero mean random variable $\mathbf{X} \sim \mathcal{X}$ and $Y \in \mathbf{R}^{N \times d_y}$ denotes the matrix consisting of samples $\{\mathbf{y}_i\}$ drawn from a zero mean random variable $\mathbf{Y} \sim \mathcal{Y}$. For notational and formulation simplicity, we assume that $d_x = d_y = d$ in the remainder of the paper although the results hold for general $d_x$ and $d_y$.

Let $C_X, C_Y$ be the covariance matrices of $\mathbf{X}, \mathbf{Y}$. Also, let $C_{XY}$ be the cross-correlation matrix between $\mathbf{X}$ and $\mathbf{Y}$. Then, we can write the CCA objective as

$$\max_{U, V} \quad F = \text{trace}\left(U^T C_{XY} V\right) \quad \text{subject to} \quad U^T C_X U = I_k \quad V^T C_Y V = I_k \qquad (2)$$

Here, $U \in \mathbf{R}^{d \times k}$ ($V \in \mathbf{R}^{d \times k}$) is the matrix consisting of $\{\mathbf{u}_j\}$ ($\{\mathbf{v}_j\}$) , where ($\{\mathbf{u}_j\}$, $\{\mathbf{v}_j\}$) are the canonical directions. The constraints in equation 2 are called *whitening constraints*.

Let us define matrices $\widetilde{U}, \widetilde{V} \in \mathbf{R}^{d \times k}$ which lie on the Stiefel manifold, $\mathsf{St}(k, d)$. Also, let $S_u, S_v \in \mathbf{R}^{k \times k}$ denote upper triangular matrices and $Q_u, Q_v \in \mathsf{SO}(k)$. We can rewrite the above equation and the constraint as follows.

**A Reformulation for CCA**

$$\max_{\substack{\widetilde{U}, \widetilde{V}, S_u, S_v, Q_u, Q_v \\ U = \widetilde{U} Q_u S_u;\; V = \widetilde{V} Q_v S_v}} \widetilde{F} = \text{trace}\left(U^T C_{XY} V\right) \tag{3a}$$

$$\text{subject to} \quad U^T C_X U = I_k;\; V^T C_Y V = I_k \tag{3b}$$

$$\widetilde{U}, \widetilde{V} \in \mathsf{St}(k, d);\; Q_u, Q_v \in \mathsf{SO}(k);\; S_u, S_v \text{ is upper triangular}$$

Here, we will maximize (3a) with respect to $\widetilde{U}$, $\widetilde{V}$, $S_u$, $S_v$, $Q_u$, and $Q_v$ satisfying equation 3b.

**Main adjustment from (2) to (3):** In (2), while $U$ and $V$ should decorrelate $C_X$ and $C_Y$ respectively, the optimization/search is unrestricted and treats them as arbitrary matrices. In contrast, equation 3 additionally decomposes $U$ and $V$ as $U = \widetilde{U} Q_u S_u$ and $V = \widetilde{V} Q_v S_v$ with the components as structured matrices. Hence, the optimization is *regularized*.

The above adjustment raises two questions: **(i)** does there exist a non-empty feasible solution set for (3)? **(ii)** if a solution to (3) can be found (which we will describe later), how "good" is this solution for the CCA objective problem, i.e., for (2)?

**Existence of a feasible solution:** We need to evaluate if the constraints in (3b) can be satisfied at all. Observe that by using $\widetilde{U}$ to be the top-$k$ principal directions of $X$, $S_u$ to be the $1/\sqrt{}$top-k eigen values of $C_X$ and $Q_u$ to be any orthogonal matrix, we can easily satisfy the "whitening constraint" and hence $\widetilde{U} Q_u S_u$ is a feasible solution of $U$ in (3) and similarly for $V$. From this starting point, we can optimize the objective while ensuring that we maintain feasibility.

**Is the solution for equation 3 a good approximation for equation 2?:** We can show that under some assumptions, the estimator for canonical correlation, i.e., solution of equation 3, is consistent, i.e., solves equation 2. We will state this formally shortly.

Before characterizing the properties of a solution for equation 3, we first provide some additional intuition behind equation 3 and describe how it helps computationally.

**Intuition behind the decomposition** $U = \tilde{U} Q_u S_u$**:** A **key** observation is the following. Recall that by performing principal component analysis (PCA), the resultant projection matrix will exactly satisfy the decorrelation condition needed for the "whitening constraint" in equation 2 (projection matrix consists of the eigen-vectors of $X^T X$). A natural question to ask is: *Can we utilize streaming PCA algorithm to help us obtain an efficient streaming CCA algorithm?* Let us assume that our estimate for canonical correlation directions, i.e., solutions of equation 3, lies in the principal subspace calculated above. If so, we can use the decomposition $U = \tilde{U} A_u$ (analogously for $V$), where $\tilde{U}$ contains the principal directions, i.e., $\in \mathsf{St}(k, d)$ and $A_u$ is a full rank $k \times k$ matrix containing the coefficients of the span. But maintaining the full rank constraint during optimization is hard, so we further decompose $A_u$ into $A_u = Q_u S_u$ with $Q_u \in \mathsf{SO}(k)$; $S_u$ is upper triangular. Additionally, we ensure the diagonal of $S_u$ to be non-zero to maintain full-rank of $S_u$. During optimization, we can maintain the non-zero diagonal entries by optimizing the $\log$ of the diagonal entries instead.

**Why equation 3 helps?** First, we note that CCA seeks to maximize the total correlation under the constraint that different components are decorrelated. The difficult part in the optimization is to ensure decorrelation, which leads to a higher complexity in existing streaming CCA algorithms. On the contrary, in equation 3, we separate equation 2 into finding the PCs and finding the linear coefficients for the span of principal directions. Then, by utilizing an efficient streaming PCA algorithm, a lower complexity can be achieved. We will defer describing the specific details of the optimization itself until the next sub-section. First, we will show formally why substituting equation 2 with equation 3a–equation 3b is sensible under some assumptions.

### 2.1 How to use the reformulation in equation 3?

We first start by stating some mild assumptions needed for the analysis.

**Assumptions: (a)** The random variables $\mathbf{X} \sim \mathcal{N}(\mathbf{0}, \Sigma_x)$ and $\mathbf{Y} \sim \mathcal{N}(\mathbf{0}, \Sigma_y)$ with $\Sigma_x \preceq cI_d$ and $\Sigma_y \preceq cI_d$ for some $c > 0$. **(b)** The samples $X$ and $Y$ drawn from $\mathcal{X}$ and $\mathcal{Y}$ respectively have zero mean. **(c)** For a given $k \leq d$, $\Sigma_x$ and $\Sigma_y$ have non-zero top-$k$ eigen values.

**A high-level solution to optimize $\widetilde{F}$ in equation 3:** Recall the following scheme which we briefly summarized earlier.

> **(a)** Initialize $\widetilde{U}, \widetilde{V} \in \mathsf{St}(k, d)$ as the top-$k$ eigen vectors of $C_X = (1/N)X^T X$ and $C_Y = (1/N)Y^T Y$ respectively; Initialize $Q_u$ and $Q_v$ to be random $\mathsf{SO}(k)$ matrices;
> **(b)** Set $S_u$ and $S_v$ to be diagonal matrices with the diagonal entries to be the square root of the inverse of the top-$k$ eigen values (to satisfy upper-triangular property);
> *Observe that with this initialization, the constraints in equation 3b are satisfied.* With a feasible solution for $U$ and $V$ in hand, we may optimize equation 3a while satisfying equation 3b. The specific details of how this is done is not critical at this point as long as we assume that a suitable numerical optimization scheme exists and can be implemented.

With the component matrices, we can construct the solution as $U = \widetilde{U}Q_u S_u$ and $V = \widetilde{V}Q_v S_v$.

**Why the solution makes sense?** We now show how the presented solution, assuming access to an effective numerical procedure, approximates the CCA problem presented in equation 2. We formally state the result in the following theorem with a sketch of proof (appendix includes the full proof) by first stating the following proposition and a definition.

**Definition 1.** *A random variable $\mathbf{X}$ is called sub-Gaussian if the norm given by $\|\mathbf{X}\|_\star :=$ $\inf \{d \geq 0 | \mathbf{E_X} [\exp (trace(X^T X)/d^2)] \leq 2\}$ is finite. Let $U \in \mathbf{R}^{d \times k}$, then $\mathbf{X}U$ is sub-Gaussian Vershynin (2017).*

**Proposition 1** (Reiß et al. (2020)). *Let $\mathbf{X}$ be a random variable which follows a sub-Gaussian distribution. Let $\widehat{X}$ be the approximation of $X \in \mathbf{R}^{N \times d}$ (samples drawn from $\mathcal{X}$) with the top-$k$ principal vectors. Let $\widetilde{C}_X$ be the covariance of $\widehat{X}$. Also, assume that $\lambda_i$ is the $i^{th}$ eigen value of $C_X$ for $i = 1, \cdots, d-1$ and $\lambda_i \geq \lambda_{i+1}$ for all $i$. Then, the PCA reconstruction error, denoted by $\mathcal{E}_k = \|X - \widehat{X}\|$ (in the Frobenius norm sense) can be upper bounded as follows*

$$\mathcal{E}_k \leq \min \left( \sqrt{2k}\|\Delta\|_2, \frac{2\|\Delta\|_2^2}{\lambda_k - \lambda_{k+1}} \right), \quad where \ \Delta = C_X - \widetilde{C}_X.$$

The aforementioned proposition suggests that the error between the data matrix $X$ and the reconstructed data matrix $\widehat{X}$ using the top-$k$ principal vectors is bounded.

Recall from (2) and (3) that the value of the CCA objective is denoted by $F$ and $\widetilde{F}$. The following theorem states that we can bound the error, $E = \|F - \widetilde{F}\|$ (proof is in the Appendix). The proof includes upper-bounding $E$ by the reconstruction error of the data projected on the principal directions using Prop. 1.

**Theorem 1.** *Using the hypothesis and assumptions above, the approximation error $E = \|F - \widetilde{F}\|$ is bounded and goes to zero while the whitening constraints in equation 3b are satisfied.*

Now, the only unresolved issue is an optimization scheme for equation 3a that keeps the constraints in equation 3b satisfied by leveraging the geometry of the feasible set.

### 2.2 How to numerically optimize (3a) satisfying constraints in (3b)?

**Overview.** We now describe how to maximize the formulation in equation 3a–equation 3b with respect to $\widetilde{U}, \widetilde{V}, Q_u, Q_v, S_u$ and $S_v$. We will first compute top-$k$ principal vectors to get $\widetilde{U}$ and $\widetilde{V}$. Then, we will use a gradient update rule to solve for $Q_u, Q_v, S_u$ and $S_v$ to improve the objective. Since all these matrices are "structured", care must be taken to ensure that the matrices *remain*

*on their respective manifolds* – which is where the geometry of the manifolds will offer desirable properties. We re-purpose a Riemannian stochastic gradient descent (RSGD) to do this, so call our algorithm *RSG+*. Of course, more sophisticated Riemannian optimization techniques can be substituted in. For instance, different Riemannian optimization methods are available in Absil et al. (2007) and optimization schemes for many manifolds are offered in PyManOpt Boumal et al. (2014).

The algorithm block is presented in Algorithm 1 (a direct implementable block for the algorithm including the expression for gradients is presented in the Appendix A.3). Let $\widetilde{F}_{\text{pri}} = \text{trace}\left(U^T C_X U\right)) + \text{trace}\left(V^T C_Y V\right))$ be the contribution from the principal directions which we used to ensure the "whitening constraint". Let $\widetilde{F}_{\text{can}} = \text{trace}\left(U^T C_{XY} V\right)$ be the contribution from the canonical correlation directions. The algorithm consists of four main blocks denoted by different colors, namely **(a)** the Red block deals with gradient calculation of the objective function where we calculate the top-$k$ principal vectors (denoted by $\widetilde{F}_{\text{pri}}$) with respect to $\widetilde{U}, \widetilde{V}$; **(b)** the Green block describes calculation of the gradient corresponding to the canonical directions (denoted by $\widetilde{F}_{\text{can}}$) with respect to $\widetilde{U}, \widetilde{V}, S_u, S_v, Q_u$ and $Q_v$; **(c)** the Gray block combines the gradient computation from both $\widetilde{F}_{\text{pri}}$ and $\widetilde{F}_{\text{can}}$ with respect to unknowns $\widetilde{U}, \widetilde{V}, S_u, S_v, Q_u$ and $Q_v$; and finally **(d)** the Blue block performs a batch update of the canonical directions $\widetilde{F}_{\text{can}}$ using Riemannian gradient updates.

**Gradient calculations.** The gradient update for $\widetilde{U}, \widetilde{V}$ is divided into two parts **(a)** The (Red block) gradient updates the "principal" directions (denoted by $\nabla_{\widetilde{U}} \widetilde{F}_{\text{pri}}$ and $\nabla_{\widetilde{V}} \widetilde{F}_{\text{pri}}$), which is specifically designed to satisfy the *whitening constraint*. Since this requires updating the principal subspaces, so, the gradient descent needs to proceed on the manifold of subspaces, i.e., on the Grassmannian. **(b)** The (green block) gradient from the objective function in equation 3, is denoted by $\nabla_{\widetilde{U}} \widetilde{F}_{\text{can}}$ and $\nabla_{\widetilde{V}} \widetilde{F}_{\text{can}}$. In order to ensure that the Riemannian gradient update for $\widetilde{U}$ and $\widetilde{V}$ stays on the manifold $\text{St}(k, d)$, we need to make sure that the gradients, i.e., $\nabla_{\widetilde{U}} \widetilde{F}_{\text{can}}$ and $\nabla_{\widetilde{V}} \widetilde{F}_{\text{can}}$ lies in the tangent space of $\text{St}(k, d)$. In order to do that, we need to first calculate the Euclidean gradient and then project on to the tangent space of $\text{St}(k, d)$.

The gradient updates for $Q_u, Q_v, S_u, S_v$ are given in the green block, denoted by $\nabla_{Q_u} \widetilde{F}_{\text{can}}, \nabla_{Q_v} \widetilde{F}_{\text{can}}$, $\nabla_{S_u} \widetilde{F}_{\text{can}}$ and $\nabla_{S_v} \widetilde{F}_{\text{can}}$. Note that unlike the previous step, this gradient only has components from canonical correlation computation. As before, this step requires first computing the Euclidean gradient and then projecting on to the tangent space of the underlying Riemannian manifolds involved, i.e., $\text{SO}(k)$ and the space of upper triangular matrices.

Finally, we get the gradient to update the canonical directions by combining the gradients which is shown in gray block. With these gradients we can perform a batch update as shown in the blue block. Using convergence results presented next in Propositions 2–3, this scheme can be shown (under some assumptions) to approximately optimize the CCA objective in equation 2.

We can now move to the convergence properties of the algorithm. We present two results stating the asymptotic proof of convergence for top-$k$ principal vectors and canonical directions in the algorithm.

**Proposition 2** (Chakraborty et al. (2020)). *(Asymptotically) If the samples, $X$, are drawn from a Gaussian distribution, then the gradient update rule presented in Step 5 in Algorithm 1 returns an orthonormal basis – the top-k principal vectors of the covariance matrix $C_X$.*

**Proposition 3.** *(Bonnabel (2013)) Consider a connected Riemannian manifold $\mathcal{M}$ with injectivity radius bounded from below by $I > 0$. Assume that the sequence of step sizes $(\gamma_l)$ satisfy the condition (a) $\sum \gamma_l^2 < \infty$ (b) $\sum \gamma_l = \infty$. Suppose $\{A_l\}$ lie in a compact set $K \subset \mathcal{M}$. We also suppose that $\exists D > 0$ such that, $g_{A_l}\left(\nabla_{A_l} \widetilde{F}, \nabla_{A_l} \widetilde{F}\right) \leq D$. Then $\nabla_{A_l} \widetilde{F} \to 0$ and $l \to \infty$.*

Notice that in our problem, the manifold $\mathcal{M}$ can be $\text{Gr}(p, n)$, $\text{St}(p, n)$ or $\text{SO}(p)$. Hence all the assumptions in Proposition 3 are satisfied if we guarantee the step sizes satisfy the aforementioned condition. One example of the step sizes that satisfies the property is $\gamma_l = \frac{1}{l+1}$.

## 2.3 Convergence rate and complexity of the RSG+ algorithm

In this section, we describe the convergence rate and complexity of the algorithm proposed in Algorithm 1. Observe that the key component of Algorithm 1 is a Riemannian gradient update. Let

$A_t$ be the generic entity needed to be updated in the algorithm using the Riemannian gradient update $A_{t+1} = \mathsf{Exp}_{A_t}\left(-\gamma_t \nabla_{A_t}\widetilde{F}\right)$, where $\gamma_t$ is the step size at time step $t$. Also assume $\{A_t\} \subset \mathcal{M}$ for a Riemannian manifold $\mathcal{M}$. The following proposition states that under certain assumptions, the Riemannian gradient update has a convergence rate of $O\left(\frac{1}{t}\right)$.

**Proposition 4.** *(Nemirovski et al. (2009); Bécigneul & Ganea (2018)) Let $\{A_t\}$ lie inside a geodesic ball of radius less than the minimum of the injectivity radius and the strong convexity radius of $\mathcal{M}$. Assume $\mathcal{M}$ to be a geodesically complete Riemannian manifold with sectional curvature lower bounded by $\kappa \leq 0$. Moreover, assume that the step size $\{\gamma_t\}$ diverges and the squared step size converges. Then, the Riemannian gradient descent update given by $A_{t+1} = \mathsf{Exp}_{A_t}\left(-\gamma_t \nabla_{A_t}\widetilde{F}\right)$ with a bounded $\nabla_{A_t}\widetilde{F}$, i.e., $\|\nabla_{A_t}\widetilde{F}\| \leq C < \infty$ for some $C \geq 0$, converges in the rate of $O\left(\frac{1}{t}\right)$.*

All Riemannian manifolds we used, i.e., $\mathsf{Gr}(k,d)$, $\mathsf{St}(k,d)$ and $\mathsf{SO}(k)$ are geodesically complete, and these manifolds have non-negative sectional curvatures, i.e., lower bounded by $\kappa = 0$. Now, as long as the Riemannian updates lie inside the geodesic ball of radius less than the minimum of injectivity and convexity radius, the convergence rate for RGD applies in our setting.

**Running time.** To evaluate time complexity, we must look at the main compute-heavy steps needed. The basic modules are $\mathsf{Exp}$ and $\mathsf{Exp}^{-1}$ maps for $\mathsf{St}(k,d)$, $\mathsf{Gr}(k,d)$ and $\mathsf{SO}(k)$ manifolds (see Table 4 in the appendix). Observe that the complexity of these modules is influenced by the complexity of svd needed for the $\mathsf{Exp}$ map for the $\mathsf{St}$ and $\mathsf{Gr}$ manifolds. Our algorithm involves structured matrices of size $d \times k$ and $k \times k$, so any matrix operation should not exceed a cost of $O(\max(d^2k, k^3))$, since in general $d \gg k$. Specifically, the most expensive calculation is SVD of matrices of size $d \times k$, which is $O(d^2k)$, see Golub & Reinsch (1971). All other calculations are dominated by this term.

## 3 EXPERIMENTS

We first evaluate RSG+ for extracting top-$k$ canonical components on three benchmark datasets and show that it performs favorably compared with Arora et al. (2017). Then, we show that RSG+ can also fits into feature learning in DeepCCA Andrew et al. (2013), and can scale to large feature dimensions where the non-stochastic method fails to. Finally we show that RSG+ can be used to improve fairness of deep neural networks without needing labels of protected attributes during training.

---

**Algorithm 1:** Riemannian SGD based algorithm (RSG+) to compute canonical directions

**Input:** $X \in \mathbf{R}^{N \times d_x}, Y \in \mathbf{R}^{N \times d_y}, k > 0$
**Output:** $U \in \mathbf{R}^{d_x \times k}, V \in \mathbf{R}^{d_y \times k}$

1 Initialize $\widetilde{U}, \widetilde{V}, Q_u, Q_v, S_u, S_v$; Partition $X, Y$ into batches of size $B$. Batch $j^{th}$ denoted by $X_j$ and $Y_j$ ;

2 **for** $j \in \left\{1, \cdots, \lfloor\frac{N}{B}\rfloor\right\}$ **do**

    **Gradient for top-$k$ principal vectors**: calculating $\nabla_{\widetilde{U}}\widetilde{F}_{\text{pri}}, \nabla_{\widetilde{V}}\widetilde{F}_{\text{pri}}$

      1. Partition $X_j$ ($Y_j$) into $L$ ($L = \lfloor\frac{B}{k}\rfloor$) blocks of size $d_x \times k$ ($d_y \times k$);

      2. Let the $l^{th}$ block be denoted by $Z_l^x$ ($Z_l^y$);

      3. Orthogonalize each block and let the orthogonalized block be denoted by $\hat{Z}_l^x$ ($\hat{Z}_l^y$);

      4. Let the subspace spanned by each $\hat{Z}_l^x$ (and $\hat{Z}_l^y$) be $\hat{\mathcal{Z}}_l^x \in \mathsf{Gr}(k, d_x)$ (and $\hat{\mathcal{Z}}_l^y \in \mathsf{Gr}(k, d_y)$);

      5. Update $\nabla_{\widetilde{U}}\widetilde{F}_{\text{pri}}$ and $\nabla_{\widetilde{V}}\widetilde{F}_{\text{pri}}$ based on $\hat{\mathcal{Z}}_l^x$ and $\hat{\mathcal{Z}}_l^y$ respectively.

3

    **Gradient from equation 3**: calculating $\nabla_{\widetilde{U}}\widetilde{F}_{\text{can}}, \nabla_{\widetilde{V}}\widetilde{F}_{\text{can}}, \nabla_{Q_u}\widetilde{F}_{\text{can}}, \nabla_{Q_v}\widetilde{F}_{\text{can}}, \nabla_{S_u}\widetilde{F}_{\text{can}}, \nabla_{S_v}\widetilde{F}_{\text{can}}$

    Calculation of the Riemannian gradients of $\widetilde{U}, \widetilde{V}, Q_u, Q_v, S_u$ and $S_v$ from equation 3, i.e., objective from CCA.

4

    **Gradient to update canonical directions**: calculating $\nabla_A\widetilde{F}$ where, $A \in \{\widetilde{U}, \widetilde{V}, Q_u, Q_v, S_u, S_v\}$

    The final gradients of $\widetilde{U}$ and $\widetilde{V}$ is a combination of gradient from objective for principal vectors and CCA; On the other hand, the gradients for $Q_u, Q_v, S_u, S_v$ is from only CCA objective;

5

    **Batch update of canonical directions**

    $A = \mathsf{Exp}_A\left(-\gamma_j \nabla_A\widetilde{F}\right)$ where, $A$ is a generic entity: $A \in \{\widetilde{U}, \widetilde{V}, Q_u, Q_v, S_u, S_v\}$;

6

7 **end**

8 $U = \widetilde{U}Q_uS_u$ and $V = \widetilde{V}Q_vS_v$;

---

## 3.1 CCA ON FIXED DATASETS

**Datasets and baseline.** We conduct experiments on three benchmark datasets (MNIST LeCun et al. (2010), Mediamill Snoek et al. (2006) and CIFAR-10 Krizhevsky (2009)) to evaluate the performance of RSG+ to extract top-$k$ canonical components. To our best knowledge, Arora et al. (2017) is the only previous work which stochastically optimizes the population objective in a streaming fashion and can extract top-$k$ components, so we compare our RSG+ with the matrix stochastic gradient (MSG) method proposed in Arora et al. (2017) (There are two methods proposed in Arora et al. (2017) and we choose MSG because it performs better in the experiments of Arora et al. (2017)). The details about the three datasets and how we process them are as follows:

**MNIST** LeCun et al. (2010): MNIST contains grey-scale images of size $28 \times 28$. We use its full training set containing 60K images. Every image is split into left/right half, which are used as the two views. **Mediamill** Snoek et al. (2006): Mediamill contains around 25.8K paired features of videos and corresponding commentary of dimension $120, 101$ respectively. **CIFAR-10** Krizhevsky (2009): CIFAR-10 contains 60K $32 \times 32$ color images. Like MNIST, we split the images into left/right half and use them as two views.

**Evaluation metric.** We choose to use Proportion of Correlations Captured (PCC) which is widely used Ma et al. (2015); Ge et al. (2016), partly due to its efficiency, especially for relatively large datasets. Let $\hat{U} \in R^{d_x \times k}, \hat{V} \in R^{d_y \times k}$ denote the estimated subspaces returned by RSG+, and $U^* \in R^{d_x \times k}, V^* \in R^{d_y \times k}$ denote the true canonical subspaces (all for top-$k$). The PCC is defined as PCC $= \frac{\text{TCC}(X\hat{U}, Y\hat{V})}{\text{TCC}(XU^*, YV^*)}$, where TCC is the sum of canonical correlations between two matrices.

**Performance.** See A.4 for the implementation deails. The performance in terms of PCC as a function of # of seen samples (coming in a streaming way) are shown in Fig. 1, and the runtime is reported in A.5 . Our RSG+ captures more correlation than MSG Arora et al. (2017) while being $5 - 10$ times faster. One case where our RSG+ underperform Arora et al. (2017) is when the top-$k$ eigenvalues are dominated by the top-$l$ eigenvalues with $l < k$ (Fig. 1b): on Mediamill dataset, the top-4 eigenvalues of the covariance matrix in view 1 are: $8.61, 2.99, 1.15, 0.37$. The first eigenvalue is dominantly large compared with the rest and our RSG+ performs better for $k = 1$ and worse than Arora et al. (2017) for $k = 2, 4$. We also plot the runtime of RSG+ under different data dimension (set $d_x = d_y = d$) and number of total samples sampled from joint gaussian distribution in A.5.

We implemented the method from Yger et al. (2012) and conduct experiments on the three datasets above. The results are shown in Table 1. We tune the step size between $[0.0001, 0.1]$ and $\beta = 0.99$ as used in their paper. On MNIST and MEDIAMILL, the method performs comparably with ours except $k = 4$ case on MNIST where it does not converge well. Since this algorithms also has a $d^3$ complexity, the runtime is $100\times$ more than ours on MNIST and $20\times$ more on Mediamill. On CIFAR10, we fail to find a suitable step size for convergence.

## 3.2 CCA FOR DEEP FEATURE LEARNING

**Background and motivation.** A deep neural network (DNN) extension of CCA was proposed by Andrew et al. (2013) and has become popular in the multi-view representation learning tasks. The idea is to learn a deep neural network as the mapping from original data space to a latent space where the canonical correlations are maximized. We refer the reader to Andrew et al. (2013) for details of the task. Since deep neural networks are usually trained using SGD on mini-batches, this requires

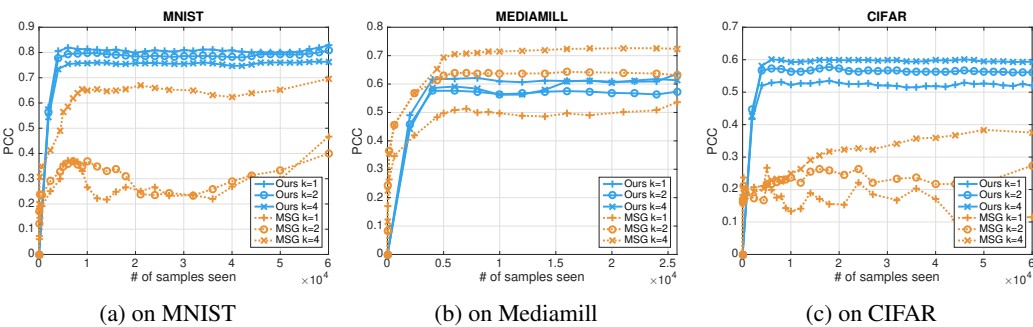

(a) on MNIST  (b) on Mediamill  (c) on CIFAR

Figure 1: Performance on three datasets in terms of PCC as a function of # of seen samples.

getting estimate of CCA objective at every iteration in a streaming fashion, thus our RSG+ can be a natural choice here. We conduct experiments on a noisy version of MNIST dataset to evaluate RSG+.

**Dataset.** We follow Wang et al. (2015a) to construct a noisy version of MNIST: View 1 is a randomly sampled image which is first rescaled to $[0, 1]$ and then rotated by a random angle from

Table 1: Results of Yger et al. (2012) (on CIFAR-10, our implementation of Yger et al. (2012) faces convergence issues).

|  | MNIST | | | Mediamill | | |
|---|---|---|---|---|---|---|
| Performance | $k = 1$ | $k = 2$ | $k = 4$ | $k = 1$ | $k = 2$ | $k = 4$ |
| PCC | 0.93 | 0.81 | 0.53 | 0.55 | 0.61 | 0.51 |
| Time (s) | 575.88 | 536.46 | 540.91 | 41.89 | 28.66 | 28.76 |

$[-\frac{\pi}{4}, \frac{\pi}{4}]$. View 2 is randomly sampled from the same class as view 1. Then we add independent uniform noise from $[0, 1]$ to each pixel. Finally the image is truncated into $[0, 1]$ to form the view 2.

**Implementation details.** We use a simple 2-layer MLP with ReLU nonlinearity, where the hidden dimension in the middle is 512 and the output feature dimension is $d \in \{100, 500, 1000\}$. After the network is trained on the CCA objective, we use a linear Support

Table 2: Results of feature learning on MNIST. N/A means fails to yield a result on our hardware.

| Accuracy(%) | $d = 100$ | $d = 500$ | $d = 1000$ |
|---|---|---|---|
| DeepCCA | 80.57 | $N/A$ | $N/A$ |
| Ours | 79.79 | 84.09 | 86.39 |

Vector Machine (SVM) to measure classification accuracy on output latent features. Andrew et al. (2013) uses the closed form CCA objective on the current batch directly, which costs $O(d^3)$ memory and time for every iteration.

**Performance.** Table 2 shows that we get similar performance when $d = 100$ and can scale to large latent dimensions $d = 1000$ while the batch method Andrew et al. (2013) encounters numerical difficulty on our GPU resources and Pytorch Paszke et al. (2019) platform in performing eigen-decomposition of $d \times d$ matrix when $d = 500$, and becomes difficult if $d$ is larger than 1000.

## 3.3 CCA for Fairness

**Background and motivation.** Fairness is becoming an important issue to consider in the design of learning algorithms. A common strategy to make an algorithm fair is to remove the influence of one/more protected attributes when training the models, see Lokhande et al. (2020). Most methods assume that the labels of protected attributes are known during training but

Table 3: Fairness results on CelebA. We applied CCA on three different layers in Resnet-18 respectively. See A.8 for positions of conv $0, 1, 2$.

|  | Accuracy(%) | DEO(%) | DDP(%) |
|---|---|---|---|
| Unconstrained | 76.3 | 22.3 | 4.8 |
| Ours-conv0 | 76.5 | 17.4 | **1.4** |
| Ours-conv1 | **77.7** | **15.3** | 3.2 |
| Ours-conv2 | 75.9 | 22.0 | 2.8 |

this may not always be possible. CCA enables considering a slightly different setting, where we may not have per-sample protected attributes which may be sensitive or hard to obtain for third-parties Price & Cohen (2019). On the other hand, we assume that a model trained to predict the protected attribute labels has been trained and is provided. For example, if the protected attribute is gender, we only assume that a well trained classifier which is trained to predict gender from the samples is available rather than sample-wise gender values themselves. We next demonstrate that fairness of the model, using standard measures, can be improved via constraints on correlation values from CCA.

**Dataset.** CelebA Wang et al. (2015b) consists of 200K celebrity face images from the internet. There are up to 40 labels, each of which is binary-valued. Here, we follow Lokhande et al. (2020) to focus on the *attactiveness* attribute (which we want to train a classifier to predict) and the *gender* is treated as "protected" since it may lead to an unfair classifier according to Lokhande et al. (2020).

**Method.** Our strategy is inspired by Morcos et al. (2018) which showed that canonical correlations can reveal the similarity in neural networks: when two networks (same architecture) are trained using different labels/schemes for example, canonical correlations can indicate how similar their features are. Our observation is the following. Consider a classifier that is trained on gender (the protected attribute), and another classifier that is trained on *attractiveness*, if the features extracted by the latter model share a high similarity with the one trained to predict gender, then it is more likely that the latter model is influenced by features in the image pertinent to gender, which will lead to an unfairly biased trained model. We show that by imposing a loss on the canonical correlation between the network being trained (but we lack per-sample protected attribute information) and a well trained classifier pre-trained on the protected attributes, we can get a more fair model. This may enable training fairer models in settings which would otherwise be difficult.

**Implementation details.** To simulate the case where we only have a pretrained network on protected attributes, we train a Resnet-18 He et al. (2016) on *gender* attribute, and when we train the classifier to predict *attractiveness*, we add a loss using the canonical correlations between these two networks on intermediate layers: $L_{\text{total}} = L_{\text{cross-entropy}} + L_{\text{CCA}}$ where the first term is the standard cross entropy term and the second term is the canonical correlation. See A.7 for more details of training/evaluation.

**Results.** We choose two commonly used error metrics for fairness: difference in Equality of Opportunity Hardt et al. (2016) (DEO), and difference in Demographic Parity Yao & Huang (2017) (DDP). See appendix A.6 for more detailed explaination of the two metrics. We conduct experiments by applying the canonical correlation loss on three different layers in Resnet-18. In Table 3, we can see that applying canonical correlation loss generally improves the DEO and DDP metrics (lower is better) over the standard model (trained using cross entropy loss only). Specifically, applying the loss on early layers like conv0 and conv1 gets better performance than applying at a relatively late layer like conv2. Another promising aspect of our approach is that is can easily handle the case where the protected attribute is a continuous variable (as long as a well trained regression network on the protected attribute is given) while other methods like Lokhande et al. (2020); Zhang et al. (2018) need to first discretize the variable and then enforce constraints which can be much more involved.

## 4 RELATED WORK

**Stochastic CCA:** There has been much interest in designing scalable and provable algorithms for CCA: Ma et al. (2015) proposed the first stochastic algorithm for CCA, while only local convergence is proven for non-stochastic version. Wang et al. (2016) designed algorithm which uses alternating SVRG combined with shift-and-invert pre-conditioning, with global convergence. These stochastic methods, together with Ge et al. (2016) Allen-Zhu & Li (2016), which reduce CCA problem to generalized eigenvalue problem and solve it by performing efficient power method, all belongs to the methods that try to solve empirical CCA problem, it can be seen as ERM approxiamtion of the priginal population objective, which requires solving numerical optimization of the empirical CCA objective on a fixed data set. These methods usually assume the access to the full dataset in the beginning, which is not very suitable for many practical applications where data tend to come in a streaming way. Recently, there are increasingly interest in considering population CCA problem Arora et al. (2017) Gao et al. (2019). The main difficulty in population setting is we have limited knowledge about the objective unless we know the distribution of $\mathbf{X}$ and $\mathbf{Y}$. Arora et al. (2017) handles this problem by deriving an estimation of gradient of population objecitve whose error can be properly bounded so that applying proximal gradient to a convex relexed objective will provably converge. Gao et al. (2019) provides tightened analysis of the time complexity of the algorithm in Wang et al. (2016), and provides sample complexity under certain distribution. The problem we are trying to solve in this work is the same as that in Arora et al. (2017); Gao et al. (2019): to optimize the population objective of CCA in a streaming fashion.

**Riemannian Optimization:** Riemannian optimization is a generalization of standard Euclidean optimization methods to smooth manifolds, which takes the following form: Given $f : \mathcal{M} \to \mathbf{R}$, solve $\min_{x \in \mathcal{M}} f(x)$, where $\mathcal{M}$ is a Riemannian manifold. One advantage is that it provides a nice way to express many constrained optimization problems as unconstrained problems. Applications include matrix and tensor factorization Ishteva et al. (2011), Tan et al. (2014), PCA Edelman et al. (1998), CCA Yger et al. (2012), and so on. Yger et al. (2012) rewrites CCA formulation as Riemannian optimization on Stiefel manifold. In our work, we further explore the ability of Riemannian optimization framework, decomposing the linear space spanned by canonical vectors into products of several matrices which lie in several different Riemannian manifolds.

## 5 CONCLUSIONS

In this work, we presented a stochastic approach (RSG+) for the CCA model based on the observation that the solution of CCA can be decomposed into a product of matrices which lie on certain structured spaces. This affords specialized numerical schemes and makes the optimization more efficient. The optimization is based on Riemannian stochastic gradient descent and we provide a proof for its $O(\frac{1}{t})$ convergence rate with $O(d^2 k)$ time complexity per iteration. In experimental evaluations, we find that our RSG+ behaves favorably relative to the baseline stochastic CCA method in capturing the correlation in the datasets. We also show the use of RSG+ in the DeepCCA setting showing feasibility when scaling to large dimensions as well as in an interesting use case in training fair models.

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

# A   APPENDIX

## A.1   A BRIEF REVIEW OF RELEVANT DIFFERENTIAL GEOMETRY CONCEPTS

To make the paper self-contained, we briefly review certain differential geometry concepts. We only include a condensed description – needed for our algorithm and analysis – and refer the interested reader to Boothby (1986) for a comprehensive and rigorous treatment of the topic.

**Riemannian Manifold:** A Riemannian manifold, $\mathcal{M}$, (of dimension $m$) is defined as a (smooth) topological space which is locally diffeomorphic to the Euclidean space $\mathbf{R}^m$. Additionally, $\mathcal{M}$ is equipped with a Riemannian metric $g$ which can be defined as $g_X : T_X\mathcal{M} \times T_X\mathcal{M} \rightarrow \mathbf{R}$, where $T_X\mathcal{M}$ is the tangent space at $X$ of $\mathcal{M}$, see Fig. 2.

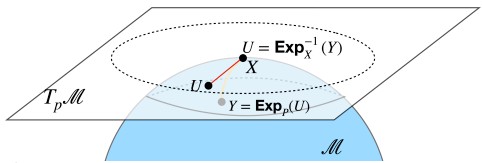

Figure 2: Schematic description of an exemplar manifold ($\mathcal{M}$) and the visual illustration of $\mathsf{Exp}$ and $\mathsf{Exp}^{-1}$ map.

If $X \in \mathcal{M}$, the Riemannian Exponential map at $X$, denoted by $\mathsf{Exp}_X : T_X\mathcal{M} \rightarrow \mathcal{M}$ is defined as $\gamma(1)$ where $\gamma : [0,1] \rightarrow \mathcal{M}$. We can find $\gamma$ by solving the following differential equation: $\gamma(0) = X, (\forall t_0 \in [0,1])\frac{d\gamma}{dt}\big|_{t=t_0} = U$. In general $\mathsf{Exp}_X$ is not invertible but the inverse $\mathsf{Exp}_X^{-1} : \mathcal{U} \subset \mathcal{M} \rightarrow T_X\mathcal{M}$ is defined only if $\mathcal{U} = \mathcal{B}_r(X)$, where $r$ is called the *injectivity radius* Boothby (1986) of $\mathcal{M}$. This concept will be useful to define the mechanics of gradient descent on the manifold.

In our reformulation, we will shortly make use of the following manifolds, specifically, when decomposing $U$ and $V$ into a product of several matrices. (a) $\mathsf{St}(p,n)$: the manifold consists of $n \times p$ column orthonormal matrices (b) $\mathsf{Gr}(p,n)$: the manifold consists of $p$-dimensional subspaces in $\mathbf{R}^n$ (c) $\mathsf{SO}(n)$, the manifold/group consists of $n \times n$ special orthogonal matrices, i.e., space of orthogonal matrices with determinant 1.

**Differential Geometry of $\mathsf{SO}(n)$:** $\mathsf{SO}(n)$ is a compact Riemannian manifold, hence by the Hopf-Rinow theorem, it is also a geodesically complete manifold Helgason (2001). Its geometry is well understood and we recall a few relevant concepts here and refer the reader to Helgason (2001) for details. $\mathsf{SO}(n)$ has a Lie group structure and the corresponding Lie algebra, $\mathfrak{so}(n)$, is defined as, $\mathfrak{so}(n) = \{W \in \mathbf{R}^{n \times n} | W^T = -W\}$. In other words, $\mathfrak{so}(n)$ (the set of Left invariant vector fields with associated Lie bracket) is the set of $n \times n$ anti-symmetric matrices. The Lie bracket, $[,]$, operator on $\mathfrak{so}(n)$ is defined as the commutator, i.e., for $U, V \in \mathfrak{so}(n)$, $[U, V] = UV - VU$. Now, we can define a Riemannian metric on $\mathsf{SO}(n)$ as follows: $\langle U, V \rangle_X = trace\left(U^T V\right)$ where, $U, V \in T_X(\mathsf{SO}(n))$, $X \in \mathsf{SO}(n)$. Note that, it can be shown that this is a bi-invariant Riemannian metric. Under this bi-invariant metric, now we define the Riemannian exponential and inverse exponential map as follows. Let, $X, Y \in \mathsf{SO}(n)$, $U \in T_X(\mathsf{SO}(n))$. Then, $Exp_X^{-1}(Y) = X \log(X^T Y)$ and $Exp_X(U) = X \exp(X^T U)$ where, exp, log are the matrix exponential and logarithm respectively.

**Differential Geometry of the Stiefel manifold:** The set of all full column rank $(n \times p)$ dimensional real matrices form a Stiefel manifold, $\mathsf{St}(p,n)$, where $n \geq p$. A compact Stiefel manifold is the set of all column orthonormal real matrices. When $p < n$, $\mathsf{St}(p,n)$ can be identified with $\mathsf{SO}(n)/SO(n-p)$. Note that, when we consider the quotient space, $\mathsf{SO}(n)/SO(n-p)$, we assume that $\mathsf{SO}(n-p) \simeq F(\mathsf{SO}(n-p))$ is a subgroup of $\mathsf{SO}(n)$, where, $F : \mathsf{SO}(n-p) \rightarrow \mathsf{SO}(n)$ defined by $X \mapsto \begin{bmatrix} I_p & 0 \\ 0 & X \end{bmatrix}$ is an isomorphism from $\mathsf{SO}(n-p)$ to $F(\mathsf{SO}(n-p))$.

**Differential Geometry of the Grassmannian $\mathsf{Gr}(p,n)$:** The Grassmann manifold (or the Grassmannian) is defined as the set of all $p$-dimensional linear subspaces in $\mathbf{R}^n$ and is denoted by $\mathsf{Gr}(p,n)$, where $p \in \mathbf{Z}^+$, $n \in \mathbf{Z}^+$, $n \geq p$. Grassmannian is a symmetric space and can be identified with the quotient space $\mathsf{SO}(n)/S(O(p) \times O(n-p))$, where $S(O(p) \times O(n-p))$ is the set of all $n \times n$ matrices whose top left $p \times p$ and bottom right $n-p \times n-p$ submatrices are orthogonal and all other entries are 0, and overall the determinant is 1. A point $\mathcal{X} \in \mathsf{Gr}(p,n)$ can be specified by a basis, $X$. We say that $\mathcal{X} = \mathrm{Col}(X)$ if $X$ is a basis of $\mathcal{X}$, where $\mathrm{Col}(.)$ is the column span operator. It is easy to see that the general linear group $\mathrm{GL}(p)$ acts isometrically, freely and properly on $\mathsf{St}(p,n)$. Moreover, $\mathsf{Gr}(p,n)$ can be identified with the quotient space $\mathsf{St}(p,n)/\mathrm{GL}(p)$. Hence, the projection

map $\Pi : \mathsf{St}(p, n) \to \mathsf{Gr}(p, n)$ is a *Riemannian submersion*, where $\Pi(X) \triangleq \mathrm{Col}(X)$. Moreover, the triplet $(\mathsf{St}(p, n), \Pi, \mathsf{Gr}(p, n))$ is a fiber bundle.

At every point $X \in \mathsf{St}(p, n)$, we can define the *vertical space*, $\mathcal{V}_X \subset T_X \mathsf{St}(p, n)$ to be $\mathrm{Ker}(\Pi_{*X})$. Further, given $g^{\mathsf{St}}$, we define the *horizontal space*, $\mathcal{H}_X$ to be the $g^{\mathsf{St}}$-orthogonal complement of $\mathcal{V}_X$. Now, from the theory of principal bundles, for every vector field $\widetilde{U}$ on $\mathsf{Gr}(p, n)$, we define the *horizontal lift* of $\widetilde{U}$ to be the unique vector field $U$ on $\mathsf{St}(p, n)$ for which $U_X \in \mathcal{H}_X$ and $\Pi_{*X} U_X = \widetilde{U}_{\Pi(X)}$, for all $X \in \mathsf{St}(p, n)$. As, $\Pi$ is a Riemannian submersion, the isomorphism $\Pi_{*X}|_{\mathcal{H}_X} : \mathcal{H}_X \to T_{\Pi(X)} \mathsf{Gr}(p, n)$ is an isometry from $(\mathcal{H}_X, g_X^{\mathsf{St}})$ to $(T_{\Pi(X)} \mathsf{Gr}(p, n), g_{\Pi(X)}^{\mathsf{Gr}})$. So, $g_{\Pi(X)}^{\mathsf{Gr}}$ is defined as:

$$g_{\Pi(X)}^{\mathsf{Gr}}(\widetilde{U}_{\Pi(X)}, \widetilde{V}_{\Pi(X)}) = g_X^{\mathsf{St}}(U_X, V_X) = \mathrm{trace}((X^T X)^{-1} U_X^T V_X) \tag{4}$$

where, $\widetilde{U}, \widetilde{V} \in T_{\Pi(X)} \mathsf{Gr}(p, n)$ and $\Pi_{*X} U_X = \widetilde{U}_{\Pi(X)}, \Pi_{*X} V_X = \widetilde{V}_{\Pi(X)}, U_X \in \mathcal{H}_X$ and $V_X \in \mathcal{H}_X$.

We covered the exponential map and the Riemannian metric above, and their explicit formulation for manifolds listed above is provided for easy reference in Table 4.

| | $g_X(U, V)$ | $\mathrm{Exp}_X(U)$ | $\mathrm{Exp}_X^{-1}(Y)$ |
|---|---|---|---|
| $\mathsf{St}(p, n)$ Kaneko et al. (2012) | $\mathrm{trace}\left(U^T V\right)$ | $\widetilde{U}\widetilde{V}^T,$ $\widetilde{U}S\widetilde{V}^T = \mathrm{svd}(X + U)$ | $(Y - X) - X(Y - X)^T X$ |
| $\mathsf{Gr}(p, n)$ Absil et al. (2004) | $\mathrm{trace}\left(\Pi_*^{-1}(U)^T \Pi_*^{-1}(V)\right)$ | $\widehat{U}\widehat{V}^T,$ $\widehat{U}\widehat{S}\widehat{V}^T = \mathrm{svd}(\bar{X} + U)$ | $\bar{Y}\left(\bar{X}^T \bar{Y}\right)^{-1} - \bar{X},$ $X = \Pi(\bar{X}), Y = \Pi(\bar{Y})$ |
| $\mathsf{SO}(n)$ Subbarao & Meer (2009) | $\mathrm{trace}\left(X^T U X^T V\right)$ | $X\,\mathrm{expm}\left(X^T U\right)$ | $X\,\mathrm{logm}\left(X^T Y\right)$ |

Table 4: Explicit forms for some operations we need. $\Pi(X)$ returns $X$'s column space; $\Pi_*$ is $\Pi$'s differential.

### A.2 Proof of Theorem 1

We first restate the assumptions from section 3.1:

**Assumptions: (a)** The random variables $\mathbf{X} \sim \mathcal{N}(\mathbf{0}, \Sigma_x)$ and $\mathbf{Y} \sim \mathcal{N}(\mathbf{0}, \Sigma_y)$ with $\Sigma_x \preceq cI_d$ and $\Sigma_y \preceq cI_d$ for some $c > 0$. **(b)** The samples $X$ and $Y$ drawn from $\mathcal{X}$ and $\mathcal{Y}$ respectively have zero mean. **(c)** For a given $k \leq d$, $\Sigma_x$ and $\Sigma_y$ have non-zero top-$k$ eigen values.

Let $F$ be the trace value solution for Eq. (2), and $\widetilde{F}$ be the trace value solution for Eqs. equation 3a, equation 3b, we next restate Theorem 1 and give its proof:

**Theorem.** *Under the assumptions and notations above, the approximation error $E = \|F - \widetilde{F}\|$ is bounded and goes to zero while the whitening constraints in equation 3b are satisfied.*

*Proof.* Let $Q_u, S_u, Q_v, S_v$ be the solutions for Eqs. equation 3a and equation 3b, $\widetilde{U}, \widetilde{V}$ be matrices consisting of top-$k$ eigen vectors of $(1/N)X^T X$ and $(1/N)Y^T Y$ respectively, $U, V$ be solutions for equation 2. Let $\widetilde{X}_u = X\widetilde{U}Q_u S_u$ and $\widetilde{Y}_v = Y\widetilde{V}Q_v S_v$. Also let $X_u = XU$ and $Y_v = YV$. Observe that mean of $X_u, Y_v, \widetilde{X}_u$ and $\widetilde{Y}_v$ are zero. Moreover the sample covariance of $X_u$ and $Y_v$ are given by $U^T C_X U$ and $V^T C_Y V$ respectively. Thus by the constraint in equation 2, $X_u^T X_u = I_k$ and $Y_v^T Y_v = I_k$. Let these covariance matrices be denoted by $C(X_u)$ and $C(Y_v)$ respectively. Analogously the sample covariance of $\widetilde{X}_u$ and $\widetilde{Y}_v$ are given by $S_u^T Q_u^T \widetilde{U}^T C_X \widetilde{U} Q_u S_u$ and $S_v^T Q_v^T \widetilde{V}^T C_Y \widetilde{U} Q_v S_v$ respectively. Let these covariance matrices be denoted by $C(\widetilde{X}_u)$ and $C(\widetilde{Y}_v)$ respectively.

Using Def. 1, we know $X_u, X_v, \widetilde{X}_u$ and $\widetilde{Y}_v$ follow sub-Gaussian distributions.

Let $F = \mathrm{trace}\left(U^T C_{XY} V\right)$ which can be rewritten as $F = \mathrm{trace}\left(X_u^T Y_v\right)$. Moreover, let $\widetilde{F} = \mathrm{trace}\left(S_u^T Q_u^T \widetilde{U}^T C_{XY} \widetilde{V} Q_v S_v\right)$ which similarly can be rewritten by $\widetilde{F} = \mathrm{trace}\left(\widetilde{X}_u^T \widetilde{Y}_v\right)$.

Consider the approximation error between the objective functions as $E = |F - \widetilde{F}|$. We can rewrite $E = |\mathrm{trace}\left(X_u^T Y_v\right) - \mathrm{trace}\left(\widetilde{X}_u^T \widetilde{Y}_v\right)|$. Due to von Neumann's trace inequality and Cauchy–Schwarz

inequality, we have

$$
\begin{aligned}
E &= |\text{trace}\left(\widetilde{X}_u^T \widetilde{Y}_v - X_u^T Y_v\right)| \\
&\leq |\text{trace}\left(\left(\widetilde{X}_u - X_u\right)^T \left(\widetilde{Y}_v - Y_v\right)\right)| \quad \text{(using Von Neumann's trace inequality)} \\
&\leq \sum_i \sigma_i(\widetilde{X}_u - X_u)\sigma_i(\widetilde{Y}_v - Y_v) \\
&\leq \|\left(\widetilde{X}_u - X_u\right)\|_F \|\left(\widetilde{Y}_v - Y_v\right)\|_F \quad \text{(using Cauchy-Scwartz's inequality)} \quad \text{(A.1)}
\end{aligned}
$$

where $\sigma_i(A)$ denote the $i$ th singular value of matrix A and $\|\bullet\|_F$ denotes the Frobenius norm.

Now, using Proposition 1, we get

$$
\|\left(\widetilde{X}_u - X_u\right)\|_F \leq \min\left(\sqrt{2k}\|\Delta_x\|_2, \frac{2\|\Delta_x\|_2^2}{\lambda_k^x - \lambda_{k+1}^x}\right)
$$

$$
\|\left(\widetilde{Y}_v - Y_v\right)\|_F \leq \min\left(\sqrt{2k}\|\Delta_y\|_2, \frac{2\|\Delta_y\|_2^2}{\lambda_k^y - \lambda_{k+1}^y}\right) \quad \text{(A.2)}
$$

where, $\Delta_x = C(X_u) - C(\widetilde{X}_u)$, $\Delta_y = C(Y_v) - C(\widetilde{Y}_v)$. Here $\lambda^x$s and $\lambda^y$s are the eigen values of $C(X_u)$ and $C(Y_v)$ respectively. Now, assume that $C(X_u) = I_k$ and $C(Y_v) = I_k$ as $X_u$ and $Y_v$ are solutions of Eq. equation 2. Furthermore assume $\lambda_k^x - \lambda_{k+1}^x \geq \Lambda$ and $\lambda_k^y - \lambda_{k+1}^y \geq \Lambda$ for some $\Lambda > 0$. Then, we can rewrite Eq. equation A.1 as

$$
E \leq \min\left(\sqrt{2k}\|I_k - C(\widetilde{X}_u)\|_2, \frac{2\|I_k - C(\widetilde{X}_u)\|_2^2}{\Lambda}\right)\min\left(\sqrt{2k}\|I_k - C(\widetilde{Y}_v)\|_2, \frac{2\|I_k - C(\widetilde{Y}_v)\|_2^2}{\Lambda}\right)
$$

And as $C(\widetilde{X}_u) \to I_k$ or $C(\widetilde{Y}_v) \to I_k$, $E \to 0$. Observe that, the limiting conditions for $C(\widetilde{X}_u)$ and $C(\widetilde{Y}_v)$ can be satisfied by the "whitening" constraint. In other words, as $C(X_u) = I_k$ and $C(Y_v) = I_k$, $C(\widetilde{X}_u)$ and $C(\widetilde{Y}_v)$ converge to $C(X_u)$ and $C(Y_v)$, the approximation error goes to zero. □

### A.3 RSG+ ALGORITHM

Here we show our algorithm with more details about the gradients in every step in Alg.2.

### A.4 IMPLEMENTATION DETAILS OF CCA ON FIXED DATASET

**Implementation details.** On all three benchmark datasets, we only passed the data once for both our RSG+ and MSG Arora et al. (2017) and we use the code from Arora et al. (2017) to produce MSG results. We conducted experiments on different dimensions of target space: $k = 1, 2, 4$. The choice of $k$ is motivated by the fact that the spectrum of the datasets decays quickly. Since our RSG+ processes data in small blocks, we let data come in mini-batches (mini-batch size was set to 100).

### A.5 RUNTIME OF RSG+ AND BASELINE METHODS

The runtime comparison of RSG+ and MSG is reported in Table 5. Our algorithm is 5–10 times faster.

We also plot the runtime of our algorithm under different data dimension (set $d_x = d_y = d$) and number of total samples sampled from joint gaussian distribution in Fig. 3.

### A.6 ERROR METRICS FOR FAIRNESS

**Equality of Opportunity (EO) Hardt et al. (2016)**: A classifier $h$ is said to satisfy EO if the prediction is independent of the protected attribute $s$ (in our experiment $s$ is a binary variable where

---

**Algorithm 2:** Riemannian SGD based algorithm (RSG+) to compute canonical directions

---

**Input:** $X \in \mathbf{R}^{N \times d_x}$, $Y \in \mathbf{R}^{N \times d_y}$, $k > 0$

**Output:** $U \in \mathbf{R}^{d_x \times k}$, $V \in \mathbf{R}^{d_y \times k}$

1 Initialize $\widetilde{U}, \widetilde{V}, Q_u, Q_v, S_u, S_v$ ;

2 Partition data $X, Y$ into batches of size $B$. Let $j^{th}$ batch be denoted by $X_j$ and $Y_j$ ;

3 **for** $j \in \left\{ 1, \cdots, \lfloor \frac{N}{B} \rfloor \right\}$ **do**

**Gradient for top-$k$ principal vectors**: calculating $\nabla_{\widetilde{U}} \widetilde{F}_{\mathrm{pri}}, \nabla_{\widetilde{V}} \widetilde{F}_{\mathrm{pri}}$

    1. Partition $X_j$ ($Y_j$) into $L$ ($L = \lfloor \frac{B}{k} \rfloor$) blocks of size $d_x \times k$ ($d_y \times k$);

    2. Let the $l^{th}$ block be denoted by $Z_l^x$ ($Z_l^y$);

    3. Orthogonalize each block and let the orthogonalized block be denoted by $\hat{Z}_l^x$ ($\hat{Z}_l^y$);

    4. Let the subspace spanned by each $\hat{Z}_l^x$ (and $\hat{Z}_l^y$) be $\hat{\mathcal{Z}}_l^x \in \mathsf{Gr}(k, d_x)$ (and $\hat{\mathcal{Z}}_l^y \in \mathsf{Gr}(k, d_y)$);

$$\nabla_{\widetilde{U}} \widetilde{F}_{\mathrm{pri}} = - \sum_l \mathsf{Exp}_{\widetilde{U}}^{-1} \left( \hat{\mathcal{Z}}_l^x \right) \quad \nabla_{\widetilde{V}} \widetilde{F}_{\mathrm{pri}} = - \sum_l \mathsf{Exp}_{\widetilde{V}}^{-1} \left( \hat{\mathcal{Z}}_l^y \right) \tag{5}$$

4

**Gradient from equation 3**: calculating $\nabla_{\widetilde{U}} \widetilde{F}_{\mathrm{can}}, \nabla_{\widetilde{V}} \widetilde{F}_{\mathrm{can}}, \nabla_{Q_u} \widetilde{F}_{\mathrm{can}}, \nabla_{Q_v} \widetilde{F}_{\mathrm{can}}, \nabla_{S_u} \widetilde{F}_{\mathrm{can}}, \nabla_{S_v} \widetilde{F}_{\mathrm{can}}$

$\nabla_{\widetilde{U}} \widetilde{F}_{\mathrm{can}} = \frac{\partial \widetilde{F}}{\partial \widetilde{U}} - \widetilde{U} \frac{\partial \widetilde{F}}{\partial \widetilde{U}}^T \widetilde{U}$ $\qquad\qquad$ $\nabla_{\widetilde{V}} \widetilde{F}_{\mathrm{can}} = \frac{\partial \widetilde{F}}{\partial \widetilde{V}} - \widetilde{V} \frac{\partial \widetilde{F}}{\partial \widetilde{V}}^T \widetilde{V}$;

$\nabla_{Q_u} \widetilde{F}_{\mathrm{can}} = \frac{\partial \widetilde{F}}{\partial Q_u} - \frac{\partial \widetilde{F}}{\partial Q_u}^T$ $\qquad\qquad$ $\nabla_{Q_v} \widetilde{F}_{\mathrm{can}} = \frac{\partial \widetilde{F}}{\partial Q_v} - \frac{\partial \widetilde{F}}{\partial Q_v}^T$;

$\nabla_{S_u} \widetilde{F}_{\mathrm{can}} = \mathsf{Upper}\left( \frac{\partial \widetilde{F}}{\partial S_u} \right)$ $\qquad\qquad$ $\nabla_{S_v} \widetilde{F}_{\mathrm{can}} = \mathsf{Upper}\left( \frac{\partial \widetilde{F}}{\partial S_v} \right)$;

Here, Upper returns the upper triangular matrix of the input matrix and $\frac{\partial \widetilde{F}}{\partial \widetilde{U}}, \frac{\partial \widetilde{F}}{\partial \widetilde{V}}, \frac{\partial \widetilde{F}}{\partial Q_u}, \frac{\partial \widetilde{F}}{\partial Q_v}, \frac{\partial \widetilde{F}}{\partial S_u}, \frac{\partial \widetilde{F}}{\partial S_v}$ give the Euclidean gradients. For completeness, the closed form expression of the gradients is,

$$\frac{\partial \widetilde{F}}{\partial \widetilde{U}} = -C_{XY} \widetilde{V} Q_v S_v Q_u^T S_u^T \qquad \frac{\partial \widetilde{F}}{\partial Q_u} = -\widetilde{U}^T C_{XY} \widetilde{V} Q_v S_v S_u^T \qquad \frac{\partial \widetilde{F}}{\partial S_u} = -Q_u^T \widetilde{U}^T C_{XY} \widetilde{V} Q_v S_v$$

$$\frac{\partial \widetilde{F}}{\partial \widetilde{V}} = -C_{YX} \widetilde{U} Q_u S_u Q_v^T S_v^T \qquad \frac{\partial \widetilde{F}}{\partial Q_v} = -\widetilde{V}^T C_{YX} \widetilde{U} Q_u S_u S_v^T \qquad \frac{\partial \widetilde{F}}{\partial S_v} = -S_u^T Q_u^T \widetilde{U}^T C_{XY} \widetilde{V} Q_v$$

$$\tag{6}$$

5

**Gradient to update canonical directions**

$\nabla_{\widetilde{U}} \widetilde{F} = \nabla_{\widetilde{U}} \widetilde{F}_{\mathrm{pri}} + \nabla_{\widetilde{U}} \widetilde{F}_{\mathrm{can}}$ $\qquad\qquad$ $\nabla_{\widetilde{V}} \widetilde{F} = \nabla_{\widetilde{V}} \widetilde{F}_{\mathrm{pri}} + \nabla_{\widetilde{V}} \widetilde{F}_{\mathrm{can}}$;

$\nabla_X \widetilde{F} = \nabla_X \widetilde{F}_{\mathrm{can}}$ where, $X$ is a generic entity: $X \in \{Q_u, Q_v, S_u, S_v\}$;

6

**Batch update of canonical directions**

$A = \mathsf{Exp}_A \left( -\gamma_j \nabla_A \widetilde{F} \right)$ where, $A$ is a generic entity: $A \in \{\widetilde{U}, \widetilde{V}, Q_u, Q_v, S_u, S_v\}$;

7

8 **end**

9 $U = \widetilde{U} Q_u S_u$ and $V = \widetilde{V} Q_v S_v$;

---

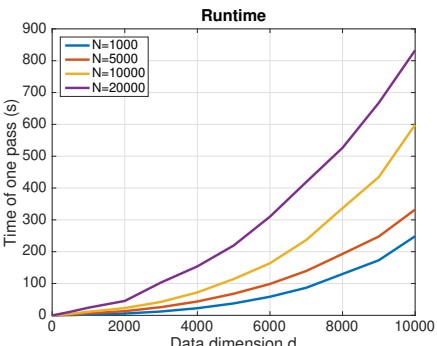

Figure 3: Runtime of RSG+ under different data dimensions and size of datasets.

Table 5: Wallclock runtime of one pass through the data of our RSG+ and MSG on MNIST, Mediamill and CIFAR (average of 5 runs).

| Time (s) | MNIST | | | Mediamill | | | CIFAR | | |
|---|---|---|---|---|---|---|---|---|---|
| | $k = 1$ | $k = 2$ | $k = 4$ | $k = 1$ | $k = 2$ | $k = 4$ | $k = 1$ | $k = 2$ | $k = 4$ |
| RSG+ (Ours) | 4.16 | 4.24 | 4.71 | 1.89 | 1.60 | 1.44 | 14.80 | 17.22 | 22.10 |
| MSG | 35.32 | 42.09 | 49.17 | 11.59 | 14.21 | 17.34 | 80.21 | 100.80 | 106.55 |

$s = 1$ stands for *Male* and $s = 0$ stands for *Female*) for classification label $y \in \{0, 1\}$. We use the difference of false negative rate (conditioned on $y = 1$) across two groups identified by protected attribute $s$ as the error metric, and we denote it as DEO.

**Demographic Parity (DP) Yao & Huang (2017)**: A classifier $h$ satisfies DP if the likelihodd of making a misclassification among the positive predictions of the classifier is independent of the protected attribute $s$. We denote the difference of demographic parity between two groups identified by the protected attribute as DDP.

## A.7 IMPLEMENTATION DETAILS OF FAIRNESS EXPERIMENTS

**Implementation details.** The network is trained for 20 epochs with learning rate $0.01$ and batch size 256. We follow Donini et al. (2018) to use NVP (novel validation procedure) to evaluate our result: first we search for hyperparameters that achieves the highest classification score and then report the performance of the model which gets minimum fairness error metrics with accuracy within the highest $90\%$ accuracies. When we apply our RSG+ on certain layers, we first use randomized projection to project the feature into 1k dimension, and then extract top-10 canonical components for training. Similar to our previous experiments on DeepCCA, the batch method does not scale to 1k dimension.

## A.8 RESNET-18 ARCHITECTURE AND POSITION OF CONV-0,1,2 IN TABLE 3

The Resnet-18 contains a first convolutional layer followed by normalization, nonlinear activation, and max pooling. Then it has four residual blocks, followed by average polling and a fully connected layer. We denote the position after the first convolutional layer as conv0, the position after the first residual block as conv1 and the position after the second residual block as conv2. We choose early layers since late layers close to the final fully connected layer will have feature that is more directly relevant to the classification variable (*attractiveness* in this case).

