# OpenReview forum: "Stochastic Canonical Correlation Analysis: A Riemannian Approach"
_ICLR.cc/2021/Conference — Reject_

### Official Review · AnonReviewer2 · 2020-10-26
**A successful approach to streaming CCA**

**Rating:** 7
**Confidence:** 3

**Review:**

The paper presents an approach to find canonical directions in a streaming fashion, i.e. without direct calculation of covariance matrices (which becomes hard when the number of examples is large). This solution to that task is not obvious, because the objective function of CCA, together with whitening constraints, does not allow simple additive decomposition.

First, the optimization task is reformulated as a task over certain Riemannian manifolds, and a natural initialization is suggested. It is shown that under a certain assumption, this initialization is already a solution of good quality. Then, a natural minimization algorithm is presented, which is based on stochastic gradient descent on a Riemannian manifold. The key aspect of the algorithm is a combination of 2 types of gradient, the gradient for top-k principal vectors and standard gradient.

The experimental part shows that the algorithm successfully solves CCA in a streaming fashion. Also, it can be effectively combined with deep feature learning (Andrew, 2013) to find common features for multi-view representation learning tasks. Experiments look convincing.

---

> ### Author Response · Authors · 2020-11-12
> **Appreciate for the positive comments**
>
> We thank the reviewer for appreciating our work. We are happy to answer any additional questions or address any outstanding concerns. Yes, we included (Andrew, 2013) in our deepCCA experiments shown in Table 1.

---

### Official Review · AnonReviewer3 · 2020-10-28
**Official Blind Review #3**

**Rating:** 6
**Confidence:** 2

**Review:**

This paper aims to reduce the computational complexity of canonical correlation analysis.By decomposing the CCA projection matrices into a product of several structured matrices, a stochastic gradient-based optimization on a Riemannian manifold is provided reducing the computational complexity from $d^3$ to $d^2k$.


Strength:
CCA is a classic and still important method, especially in combination with deep neural networks (e.g., multi-view learnings).
The proposed method enables the applications of CCA to high-dimensional vectors with small memory.
This makes it easier to use CCA as an objective function of deep neural networks, which is trained on GPUs.
Experiments show the benefits of their proposed method, in particular, the computational speed is 5-10 times faster than the existing method (MSG).

Weakness:
Although the authors claim that their proposed method captures more correlation than MSG, two of the three datasets in which their method is superior (MNIST and CIFAR) are not realistic setting (i.e., correlation between left/right half of images).

Question:
Is it possible to make a (numerical) comparison with Yger+2012, which reformulates CCA as an optimization on the generalized Stiefel manifold?

---

> ### Author Response · Authors · 2020-11-13
> **Experimental clarification+addtional comparison**
>
> 1. Claim that their proposed method captures more correlation than MSG, two of the three datasets in which their method is superior (MNIST and CIFAR) are not realistic settings (i.e., correlation between left/right half of images).
>
> Ans: We chose these datasets mainly because they are commonly used test-beds in other papers which study the CCA model (such as Ge et al. 2016 and Andrew et al. 2013). At a minimum,  they help in evaluating whether a model works satisfactorily in settings where one would certainly expect sufficient correlation. The use of CCA for the fairness experiment was designed to leverage the model’s ability to work in a high dimensional setting, which will otherwise present a key bottleneck. Other use cases which could benefit from a CCA type objective during the training process, appear to be feasible.
>
> 2. Is it possible to make a (numerical) comparison with Yger+2012, which reformulates CCA as an optimization on the generalized Stiefel manifold?
>
> Ans: We thank the reviewer for the suggestion. Yes, we will shortly update the paper with these experiments.

---

> > ### Author Response · Authors · 2020-11-15
> > **Comparison with Yger+2012**
> >
> > As per the reviewer's suggestion, we have added a comparison with Yager, 2012 in Table 1.

---

### Official Review · AnonReviewer1 · 2020-10-28
**The major concern is clarity**

**Rating:** 4
**Confidence:** 4

**Review:**

main contribution

This work offers a stochastic CCA algorithm based on Riemannian optimization approach

Strength

- The paper offers a theory-backed algorithm for CCA under the assumption that the two views are sub-Gaussian. The complexity-accuracy tradeoff of the algorithm seems to be appealing according to the experiments.

Weakness

Overall, the biggest concern is readability. The paper is very packed and many treatments seem to be unbalanced. Important details are missing, and proofs seem to be hastily. The paper may need some re-packaging and re-organization before its core technical contents could be easily followed.

- readability. The biggest concern of the work is that it is very hard to read. This creates a lot of barriers in understanding key aspects of the paper, e.g., contributions, formulations, novelty of the proof, just to name a few. The key formulation and the definitions of the manifolds were not clearly defined. The equivalence of (1) and (3) was not clearly shown. The theorems were presented in a bit abrupt way. Even the algorithm does not appear in the maintext but the appendix (which is also hard to read). The color code is used in a way that is a bit confusing (does ICLR allow writing in different colors?).

- Clarify about the contribution. It is hard to clearly see how much is the contribution. The proofs seem to be short and most of the proofs are presented using “propositions” cited from existing papers. If the authors think the contributions lie in reformulating the CCA problem as a PCA problem, then the reformulating part is perhaps the contribution. But from the current writing it is hard to follow how the reformulation comes through and how the reformulation enables using these existing propositions to prove convergence of the proposed algorithm.

- It is unclear why the reformulation in (3) is a good approximation for CCA. Some attempts for justifying this were offered in Theorem 1, but it was based on the assumption X and Y are sub-Gaussian, which is at best a special case, even if the proof is correct (which, due to the current organization of the paper, is hard to read and fully understand).

- “SO(n): group/manifold of nxn special orthogonal matrices.” what is the definition of “special” here? are they the commonly understood orthogonal matrices?

- The paper has this upper triangular structure of the S matrices but this point seems to have no detailed explanation. If one understands U = \tilde{U}S as the QR decomposition, then indeed S is upper triangular, but why is there a Q in the middle? Why is this Q useful?

- the “high-level” description of the algorithm says the idea is to connect CCA with PCA using this reformulation, but this point was not clearly explained.

- Complexity. From the current writing, it is very hard to see how the complexity of the algorithm is calculated. The gradients needed are tabulated in the supplementary materials, but it is very hard for a reader to directly see why the algorithm saves computational complexity.

- It is also unclear how the convergence and convergence rate analyses come together. These parts may need to be elaborated.

---

> ### Author Response · Authors · 2020-11-13
> **Clarifications for Reviewer 1**
>
>   1. *Paper is packed*
>
>   We hope that the reviewer agrees that most papers that draw upon a few different areas to derive the algorithm or to analyze its properties may appear to be packed. Indeed, some of the concepts can only be reviewed briefly given the page limits. This is not necessarily due to sloppy presentation.
>
>   2. *Improve readability: contributions, formulations, etc*
>
>   We request the reviewer to briefly look at page 3 again, if possible. Putting the analysis components aside temporarily, we are happy to clarify anything specific on page 3 regarding the contribution/formulation that is unclear.
>
>   Page 3 describes what the overall model is (shown in 2), what a minor adjustment to it yields (in 3a--3b) and also provides an intuition about why this may potentially help. The subsequent sections actualize this intuition and show that this is reasonable by describing the analysis and the mechanics of the numerical scheme.
>
>   3. *Equivalence of (1) and (3) was not clearly shown.*
>
>   We should note that (1) and (2) are both standard for CCA. So we can focus on (3). The reviewer can check that the adjustment from (2) -- (3) is actually minor: everything from (2) stays the same in (3) except that we further tease out the structure in $U$ as $U = \tilde{U}A$ where $A = QS$, similarly for $V$. Q10 below will help easily clarify this doubt completely.
>
>   4. *Theorems presented in a bit abrupt way.*
>
>   We will much appreciate any guidance on a segue to Theorem 1 that will help address this concern.
>
>   5. *Even the algorithm does not appear in the main text.*
>
>   Alg. 1 in the main paper is indeed the full pseudocode. The appendix simply writes out the low-level details of the gradient calculations, needed only if someone wants to cross-reference with our code. We thought that an explicit description of these calculations (tedious but not difficult) will only add marginal value and impact readability.
>
>   6. *Color code is confusing.*
>
>   The colors were used to easily reference the description in Section 2.2 with the algorithm blocks in Algorithm 1. We are happy to modify it.
>
>   7. *Main Contribution*
>
>   We hope that the clarifications here and the other reviews will help resolve this concern fully.
>
>   8. *Unclear why the reformulation in (3) is a good approximation for CCA.*
>
>   We can clarify this doubt in two parts.
>   First notice that (3) rewrites the standard CCA in (2) using the additional structure on U and V (see Q10 below). Now, the second part deals with whether this is a sensible approximation. This is where we can check Theorem 1: we show that under some assumptions, solution of (3) is a consistent estimator of the CC directions. For the non-Gaussian distribution, one can resort to a case by case analysis but will involve additional assumptions as well as  adjustments to the algorithm making a concise presentation of the ideas difficult.
>
>   9. *“SO(n)”: what is  “special” here?*
>
>   $SO(n)$ is the group of $n\times n$ orthogonal matrices with determinant 1. Referring to it as the special orthogonal group is common practice (https://mathworld.wolfram.com/SpecialOrthogonalGroup.html).
>
>   10. *Why is there a Q in the middle?*
>
>   Here $\tilde{U}$ is the matrix of principal vectors and $U$ is the canonical  directions (similarly for $V$).
>   A key observation is that $U$ lies in the column span of $\tilde{U}$, i.e., $U = \tilde{U}A$ where we have a full rank matrix $A$. But optimization on the space of full rank matrices is more challenging since we need to satisfy the rank constraint. So, we may decompose $A=QS$ where $Q$ is orthogonal and $S$ is an upper triangular matrix. Substituting $QS$ by $A$, we get $U = \tilde{U} QS$. We hope this addresses the doubt regarding (3a)-(3b).
>
>   This adjustment makes the optimization convenient. Constraints for orthogonal and upper triangular matrices can be implicitly preserved throughout the Riemannian gradient descent optimization without any explicit regularization.
>
>   11. *How high-level description connects CCA with PCA using this reformulation*
>
>   The key premise (see Reviews 2, 4) is that by treating the PCA and maximizing canonical correlation objective separately, we can achieve potential benefits. The PCA part enforces the whitening constraint. The CC directions lie in the principal subspaces, hence with an efficient scheme to compute principal directions we need to find the appropriate coefficients, which is enabled by (3).
>
>   12. *Hard to see how the complexity of algorithm is calculated.*
>
>   Our algorithm involves structured matrices of size $d\times k$ and $k\times k$, so any matrix operation should not exceed a cost of $O(\max(d^2k, k^3))$, since in general $d \gg k$, which directly gives $O(d^2k)$. Specifically, as mentioned in the paper, the most expensive calculation is SVD of matrices of size $d\times k$, which is $O(d^2k)$ (Cline and Dhilon, Handbook of Linear Algebra, 2006). All other calculations are dominated by this term.

---

> > ### Author Response · Authors · 2020-11-13
> > **Paper will be updated shortly**
> >
> > Dear Reviewer 1,
> >
> > If these clarifications are useful, please let us know and we will update the paper. We also appreciate any other suggestions.
> >
> > Thanks again for your time.

---

> > > ### Comment · AnonReviewer1 · 2020-11-13
> > > **Comments on the authors repsonse**
> > >
> > > Comment:   1. *paper is packed*
> > >
> > > The reviewer understands page limitation may give challenges for preparing the manuscript. But the clarity bar shouldn’t be lowered because authors have a lot of materials to present.
> > >
> > > 2. *Improve readability: contributions, formulations, etc*
> > >
> > > The reviewer has taken a look at page 3 again per the authors’ request. This page is particular hard to follow. First, the reviewer does not think SO, Gr, and St are clearly defined. If the authors are using some conventionally used notations from somewhere else, one standard way is to cite the origin. A better way is perhaps to give the exact definitions in the appendix. Saying SO is a manifold of “special matrices” does not help clarify what is the mathematical definition---and the argument that this is “standard” in certain community may not help neither. Without definitions of these manifolds, the reviewer could only guess what the authors meant to write here.
> > >
> > > Besides notation discrepancy, page 3 also have many comments that does not lead to clear understanding.
> > >
> > > For example. In Remark 1
> > >
> > > it seems that U, V are constrained in equation (2) rather than "arbitrary" matrices.  The decomposition adds more structural constraints on to U, V, but the remark did not explain why this structural constraint is a good constraint. Is it without loss of generality, or it is a compromise for computational efficiency? It may be helpful to clarify these points.
> > >
> > > This is perhaps related to Theorem 1. Theorem 1 only says that E goes to zero when Eq 3b is satisfied. But Theorem 1 does not say when Eq. 3b can be satisfied. If the reviewer understands correctly, the manifold parameterization for U and V may make Eq 3b infeasible, since the search space has been shrunk. This was not quite articulated in the paper. More importantly, if Eq 3b is not satisfied, then the computed U, V may not be desired solutions. How to deal with this situation?
> > >
> > > For the paragraph `` Intuition behind the decomposition''
> > >
> > > The last couple of sentences may be the most useful. But this claim was not clearly seen at this stage.
> > >
> > >    3. *Equivalence of (1) and (3) was not clearly shown.*
> > >
> > > The expression change is indeed minor. But how large is the change in essence? The reviewer’s concern is that this decomposition may have changed the problem to an extent that is not easy to quantify. The question is how much the problem has been changed?Is the approximation good?
> > >
> > > 4. *Theorems presented in a bit abrupt way.*
> > >
> > > Some suggestions: Perhaps an equivalence lemma could appear after eq 3. This may be part of Theorem 1 or modified version of Theorem 1. Proposition 1 may be a bridge, and may not need to appear in the main text. Proposition 2 may be more clearly explained how it is used to compute Eq 3, rather than just stating it here without too much explanation. The same applies to Prop 5. How is this used in *this* submission? This is perhaps more important than simply stating this proposition.
> > >
> > >   5. *Even the algorithm does not appear in the main text.*
> > >
> > > The reviewer does not think readers could follow the “full” pseudocode to reproduce the algorithm. Indeed, detailed explanations do not add too much value. But vague description may not either.
> > >
> > > 6. *Color code is confusing.*
> > >
> > > This may be just the reviewer’s personal opinion. The authors may keep it. It does not affect the score that the reviewer gives.
> > >
> > > 7. *Main Contribution*
> > >
> > > Unfortunately, the reviewer still feels the paper hard to follow. Some major re-organization may help improve the clarification level.   The proof of Theorem 1 may be enhanced to justify the reformulation.
> > >
> > >  8. *Unclear why the reformulation in (3) is a good approximation for CCA.*
> > >
> > > The reviewer has checked the proof of Theorem 1. Two major concerns. First, again, this proof is only for sub-Gaussian data, but CCA is a generic tool that can be applied as a deterministic method. Second, C(\tilde{X}_k) \rightarrow I_k is unjustified. Can it be I_k? how far is it from I_k? Assuming it approaches I_k is not “consistency proof”.
> > >
> > > 9. *“SO(n)”: what is  “special” here?*
> > >
> > > See the reviewer’s comment under Q2
> > >
> > > 10. *Why is there a Q in the middle?*
> > >
> > > The above explanation makes sense. The Q matrix is automatically full rank. The unclear part is how the upper triangular structure is preserved in the optimization? In addition, even if the upper triangular structure is preserved, S can still be rank deficient. So overall A=QS may still be rank deficient. How is this addressed?
> > >
> > > 11. *How high-level description connects CCA with PCA using this reformulation*
> > >
> > > see comments on Theorem 1

---

> > > > ### Author Response · Authors · 2020-11-15
> > > > **Clarifications for Reviewer 1**
> > > >
> > > > 1) Clarity
> > > > Ans: The reviewer’s suggestions have helped us slow down the presentation and provide more details in many places throughout the paper with major reorganization. Several steps and concepts which we assumed that a reader will already be familiar with, are now more explicitly described.
> > > >
> > > > Readability
> > > >
> > > > 2.1) give the exact definitions in the appendix
> > > > Ans: We significantly expanded the appendix describing the material to introduce the reader to every manifold we have used. We also provided additional references for the interested reader.
> > > >
> > > > 2.2) page 3 not clear
> > > > Ans: Pages 3-4 have been significantly expanded as a response to this comment.
> > > >
> > > > 2.3) why structural constraint is good
> > > > Ans: In this revised version, we devoted effort and space into describing most aspects of this adjustment and its ramifications regarding the feasibility set and computational efficiency.
> > > >
> > > > 2.4) Theorem 1 says E goes to zero when Eq 3b is satisfied.
> > > > Ans: The additional discussion related to the initialization followed by the RGD scheme (which is used specifically to maintain feasibility) on pages 2, 3, and 4, we believe will clarify these doubts.
> > > >
> > > > 2.5) If Eq 3b is not satisfied, U, V may not be desired solutions.
> > > > Ans: We can clarify this doubt. Equation (3b) can be easily satisfied by choosing $\widetilde{U}$ as the principal vectors, $S_u$ to be the top-k eigenvalues of $X^TX$ and $Q_u$ to be any special orthogonal matrix. This ensures that the initialization is feasible. Notice that the solution space of (3) is a subset of the solution space of (2) under the distributional assumption, so any feasible solution to (3) satisfies (2).
> > > >
> > > > 2.6) claims were not clear
> > > > Ans: Based on the reviewer’s suggestion, we rewrote and restructured most parts of page 3 including the part preceding and following the “Intuition” subsection, together with other improvements on pages 2 and 4/5.
> > > >
> > > >  3) Equivalence of (1), (3).
> > > > Ans: We added a sub-section specifically to describe the feasible solution set for (3). We also devoted significantly more text to describe the conditions under which our proposed CC estimator as a solution for (3) is consistent.
> > > >
> > > > 4) Thm. presented in abrupt way.
> > > > Ans: Based on these suggestions, we removed Proposition 1 and used Proposition 2 (Proposition 1 in the revised pdf) as a bridge between CC estimator and Theorem 1. We added significantly more text and explanation so that a reader will follow the reasoning much more easily.
> > > >
> > > > 5) Prop 5. how is this used
> > > > Ans: We have added text after Prop 5 to clarify this point.
> > > >
> > > >  6) algorithm isn't in main text.
> > > > Ans: We included more details of the algorithm in the text and more explanation of each of the main parts of the algorithm which will simplify understanding.
> > > >
> > > > 7) major reorganization.
> > > > Ans: This concern is now addressed in this revision. As per reviewer’s suggestion, we much improved the level of detail/explanation provided for each step and expanded the proof of Thm. 1 with additional description and annotation.
> > > >
> > > > 8) Thm. 1: sub-Gaussian data; how far from I_k
> > > > Ans: For data that severely violates the assumption, our algorithm can be applied but evaluation will be empirical. But the consistency proof may not hold or will need to be reworked or additional assumptions or algorithmic adjustments will be needed on a case-by-case basis based on the distribution at hand to make the current steps go through. While removing such assumptions is desirable, such an assumption is not uncommon for such analyses for CCA as well as many other generic models. For example, Vershynin et al. (The Mathematics of Data, 2017) notes “Sub-gaussian distributions form a sufficiently wide class of distributions. Many results in probability and data science are proved nowadays for sub-gaussian random variables…”.
> > > > Thm. 1 states that if the data is sub-Gaussian, our proposed CC estimator (soln. of (3)) is consistent, i.e., asymptotically converges to the soln. of (2).
> > > > Observe that, $C(X_k) = I_k$ using the ``whitening constraint’’ as X_k is the projection of X on U, thus  $C(\tilde{X}_k) \rightarrow I_k$ implies $C(\tilde{X}_k) \rightarrow C(X_k)$. We clarify this in the proof of Theorem 1. Observe that, with $\widetilde{U}$ as the top-k principal directions, $S_u$ as inverse of the square root of top-k eigenvalues of $X^TX$ and $Q_u$ as any SO matrix, we can satisfy $C(\tilde{X}_k)  = I_k$. But in practice, because of streaming PCA,  $C(\tilde{X}_k)  \rightarrow I_k$ asymptotically. Moreover, as $C(\tilde{X}_k)$ converges to $I_k$, the error between $F$ (objective function (1)) and $\widetilde{F}$ (objective function (3)) converges to $0$.
> > > >
> > > > 9) Even if the upper triangular structure is preserved, S can be rank deficient
> > > > Ans: We ensure the full rank of the upper triangular matrix by ensuring that the diagonal to be non-zero. During optimization, we ensure the non-zero diagonal entries by optimizing over the log of the diagonal entries. We have added additional text which describe these details.

---

> > > > > ### Comment · AnonReviewer1 · 2020-11-18
> > > > > **Some clarification regarding item 8 and item 9**
> > > > >
> > > > > There are some points that the authors may be able to quickly clarify:
> > > > >
> > > > > 1) by `"asymptotically converge": does it mean when the number of samples goes to infinity?
> > > > >
> > > > > 2) by "ensuring" the diagonal of S to be nonzero, what exactly does it mean?. A constraint such as S_ii \neq 0 may not be nontrivial in practice (and nonconvex in fact). How is this enforced without affecting the convergence proof? By optimizing over the log, you also need some nonnegativity constraint on S_ii, I suppose---otherwise log is not defined. The reviewer is wondering are you using the log-barrier approach? Not that log barrier works for convex sets, but S_ii\neq 0 is not convex. Even if the set is convex, the barrier parameter affects the approximation accuracy a lot. It is unclear how this approach affects the overall analysis since this information was not disclosed before.  In addition, your algorithm descriptions in the main text and in the supplemental material seem not to reflect this point.  The reviewer feels that this explanation seems not to be very convincing.

---

> > > > > > ### Author Response · Authors · 2020-11-18
> > > > > > **Clarification of two points**
> > > > > >
> > > > > > 1. by `"asymptotically converge": does it mean when the number of samples goes to infinity?
> > > > > >
> > > > > > Ans: Here “asymptotic” is with respect to the number of steps of the Riemannian gradient  descent procedure (Bonnabel, 2013 at https://arxiv.org/pdf/1111.5280.pdf), see (4) and Theorem 1. This analysis style has also been used in (http://proceedings.mlr.press/v97/kasai19a/kasai19a.pdf; ICML 19, see Theorem. 4.4 and Corollary 4.5)
> > > > > > (https://www.di.ens.fr/~fbach/colt_2018_tripurareni_flammarion_bach_jordan.pdf; JMLR 18, see Theorem 1) and others.
> > > > > > In our case, as the number of steps goes to infinity, $\widetilde{U}$ converges to the principal directions, and hence $C(\widetilde{X}_k)$ converges to $C(X_k)$. Note that as the number of samples is fixed, the number of steps corresponds to the number of iterations over the dataset. In our setup, we used one pass over the dataset as the setting is stochastic.
> > > > > >
> > > > > >
> > > > > > 2. By "ensuring" the diagonal of S to be nonzero, what exactly does it mean?. A constraint such as $S_{ii} \neq 0$ may not be nontrivial. How is this enforced without affecting the convergence proof? By optimizing over the log, you also need some non-negativity constraint on $S_ii$, I suppose---otherwise log is not defined. In addition, your algorithm descriptions in the main text and in the supplemental material seem not to reflect this point. The reviewer feels that this explanation seems not to be very convincing.
> > > > > >
> > > > > > Ans: Observe that, without loss of generality we can assume diagonal of $S$, i.e., $S_{ii} > 0$ as if for any $j$, $S_{jj} <0$, we can flip the sign of the $j^{th}$ column of $Q$, where, $Q$ is an orthogonal matrix. Moreover, sign flip does not affect the converge analysis.
> > > > > > Hence, $S \in \mathbf{R}^{k\times k}$ can be modeled as the manifold $\mathbf{R}^{k(k-1)/2)}\times \mathbf{R}_+^k$ ($\mathbf{R}_+$ is the space of positive reals). Now, parameterizing $\mathbf{R}_+$ can be accomplished using Riemannian log map (which is the natural logarithm in this case) (see Cheng et al., AISTATS 2011).  Observe that the log space is unrestricted, as $x > 0 \implies log(x) \in \mathbf{R}$, thus the optimization on the log space is unconstrained.
> > > > > > Further, Proposition 3 is used while optimizing for $S$, hence the convergence analysis remains valid.

---

### Official Review · AnonReviewer4 · 2020-11-01
**A stochastic linear CCA method for high dimensional data**

**Rating:** 6
**Confidence:** 4

**Review:**

1. Paper summary:

This paper proposes a method for solving linear CCA on high dimensional data. Linear CCA has a closed form solution. The solution requires a whitening step that costs O(d^3). This makes it not applicable to data in high dimensional spaces, e.g. representations learnt by deep networks.

To resolve the issue, the authors propose a reformulation of linear CCA which decomposes the transformation matrix U (V) into a product of three matrices. Those three matrices have the following properties:

- Their initial values can be obtained by efficient streaming PCA on original view matrix X (Y). Streaming PCA costs O(d^2 * k) only where k is the top k eigenvectors.

- They allow for Riemannian stochastic gradient descent which ensures their updated values lie on the same manifold.

2. Strong points of the paper:

The new linear CCA formulation justifies the rationale of batch CCA training.

Under Gaussian distribution assumption:
- The absolute difference between correlation found by original linear CCA and stochastic one is bounded.
- The convergence of the training process is proven.

The experiments are performed on different aspects:
- Recovering groundtruth transformations on MNIST, CIFAR and Mediamill data sets.
- Learning deep features on MNIST data set.
- Improving fairness in deep learning by adding CCA term to the loss function.

3. Weak points of the paper:

The theoretical results are obtained under a strong assumption that X and Y both have Gaussian distribution.

Most propositions are from other papers.

---

> ### Author Response · Authors · 2020-11-12
> **Gaussian assumption**
>
> We thank the reviewer for the review and appreciating our work. We clarify the main questions below,
>
> Q: The theoretical results are obtained under a strong assumption that X and Y both have Gaussian distribution.
>
> Ans:
> We answer in two parts,
> (1) Our setting closely followed the formulation described in a well known result on probabilistic CCA (A probabilistic interpretation of canonical correlation analysis,(Bach and Jordan, 2005 Tech report). Our convergence theorem (Theorem 1) holds under identical assumptions on $X$ and $Y$ as described in probabilistic CCA. The reviewer will likely agree that this assumption is common, even in the batch setting of CCA, to derive convergence or consistency guarantees in the finite sample regime. If we inspect the analysis described in Appendix A.2 that accompanies the result in Prop 2, we see that the steps will carry through as long as $\Delta$ is bounded. On a case by case basis, depending on the distributional assumption, some other assumptions will need to be imposed to ensure that this condition holds so that the analysis leads to the desired guarantees.
>
> (2) The key computational advantages that the algorithm offers, as noted by the reviewer, emerge from breaking down the CCA objective into PCA (which satisfies the whitening constraint) and the module which maximizes correlation. This modification leads to the computational complexity benefits. On the other hand, the consistency of our estimator to compute canonical directions requires consistency of the PC estimator as well. It is for this reason that the Gaussian assumption seemed sensible.
>
>
> Q: Most propositions are from other papers.
>
> Ans: The main theorem (Theorem 1) is original to this paper while we do utilize or restate results from other articles, as needed, at various places in the analysis.

---

### Decision · Program_Chairs · 2021-01-07
**Final Decision**

**Decision:**

Reject

**Comment:**

This paper gives a new algorithm for the CCA problem. The main idea of the new algorithm is to reformulate the matrices in the CCA problem as a product of three matrices: one orthonormal matrix, one rotation and one upper-diagonal matrix. The algorithm then performs remannian gradient descent to these components. The per-iteration complexity of the algorithm is O(d^2k) while the (local) convergence rate is O(1/t). Overall the reformulation is interesting and the algorithm seems effective in practice. On the other hand the convergence rate proof relies on local strong convexity and it's not clear why the algorithm converges globally (or even what is the radius of convergence locally).